# Concussion in Parasport: A Narrative Review of Research Published since the Concussion in Para Sport (CIPS) Group Statement (2021)

**DOI:** 10.3390/healthcare12161562

**Published:** 2024-08-07

**Authors:** Tansy Ryan, Lisa Ryan, Ed Daly

**Affiliations:** Department of Sport, Exercise and Nutrition, Atlantic Technological University, H91 T8NW Galway, Irelandlisa.ryan@atu.ie (L.R.)

**Keywords:** concussion, parasport, disability, brain injury, position statement, para-athletes

## Abstract

Sports-related concussion (SRC) is an injury whereby impact to the face/head/neck impairs cognitive functioning. Parasport athletes have an increased risk for SRC. The Previous Concussion in Sport Group iterations lack guidance for parasport SRC assessment, management, and return-to-play. This article aims to investigate the research relating to parasport SRCs published since the 2021 Concussion in Para Sport (CIPS) position statement and highlights possible new recommendations. A literature review of parasport concussions was conducted. Articles citing the 2021 publication and/or published since then were reviewed. Relevant data were extracted and discussed in this article. Since 2021, twelve (n = 12) articles emerged investigating parasport SRC. Parasport athletes experience greater concussion symptoms and severity scores compared to able-bodied athletes. Visually impaired athletes account for >50% of parasport SRCs. Wheelchair basketball and rugby have the highest SRC incidence rates across parasports. Current SRC assessment methodologies are not designed with consideration of parasport athletes’ unique experiences. Guidelines lack a return-to-learning protocol, making returning to education/work challenging for such athletes. Understanding these athletes’ SRC experiences is paramount in supporting their recovery. Specific guidelines for SRC assessment, management, return-to-play, and return-to-learn for parasport athletes are necessary to enhance their rehabilitation and avoid the occurrence of long-term symptoms.

## 1. Introduction

Sports-related concussion (SRC) is regarded as a common sports injury amongst certain parasport athletes, and in many instances, SRC in parasport has a similar incidence rate when compared to able-bodied athletes. In a 52-week prospective longitudinal cohort study collecting weekly data on the incidence proportion and rate of SRC among parasport athletes, 9.3% of parasport athletes reported experiencing at least one SRC during the one-year period. The incidence rate was 0.5 SRC per 1000 h of sporting exposure. Comparably, able-bodied sports research has reported athlete incidence rates ranging from 0.1–3.07 SRC per 1000 h of sporting exposure [1]. Regardless of being a parasport or an able-bodied athlete, there are common mechanisms of injury whereby a direct or indirect impact to the face, head, or neck may result in impaired brain or cognitive functioning. In many cases, they are oftentimes transient, but there remains a possibility for chronic long-term post-concussion symptoms to arise [2,3].

One of the primary issues with respect to SRCs is a lack of consistency in concussion reporting and the variance of data regarding SRCs [4]. Research on SRCs among able-bodies athletes has had significantly more research contributions in comparison to those investigating SRC among parasport athletes. This paucity of research in the area was highlighted by Teodoro et al. [5], who conducted literature searches relating to the topic. Their investigations returned a total of 7960 responses for the general search of SRC in sport; however, only 54 results were related to SRCs in parasport.

Previous iterations of the Concussion in Sport Group (CISG) have largely disregarded parasport athletes, failing to outline specific protocols for those with impairments. The most recent iteration from the Amsterdam CISG meeting offered some guidance and undated previous CISG statements [6]. The Amsterdam statement acknowledged that parasport athletes are at an increased risk of SRC and highlighted the need for more research involving this cohort. It was noted that the focus of future research should include an acknowledgment of parasport athlete experiences of SRCs. This was viewed as an essential component to ensure adequate support is available throughout their assessment, management, and return to sport (RTS) [6]. Issues arise as present concussion tools such as the Sports Concussion Assessment Tool 6 (SCAT6) may have limited use for parasport athletes and may not have been validated or may not be appropriate for the para-athletic population [7,8,9].

SRC in parasport athletes is a complex issue due to the many different impairment types within each classification and the variation in how they impact playing different sports [10,11]. As a result of the sparsity of information and guidance available to support concussion management in parasports, the Concussion in Para Sport (CIPS) multidisciplinary expert group drafted a detailed consensus statement. Through completing an in-depth investigation and analysis of existing research, Weiler et al. [8] published the CIPS first position statement outlining the best practices for SRC assessment, management, and return-to-sport (RTS) protocol, as well as additional considerations for parasport athletes. Since this position statement was published in 2021, numerous research articles have emerged investigating parasport athlete SRCs. This is a critical aspect for understanding and ensuring that clinicians, coaches, and support staff have the ability to support parasport athletes throughout an incidence of SRC adequately. As such, athletes oftentimes have unique needs, and “normal” baseline functions in SRC management cannot be assumed; an individualised, case-by-case approach is required [12]. It is important to note that novel approaches for supporting the cognitive functioning of those who have experienced a traumatic brain injury are emerging and seem promising. However, such research remains focused on able-bodied individuals and has yet to recruit and investigate such relationships for parasport athletes and their SRC recovery [13].

The aim of this research was to collate and investigate the research relating to SRCs in parasport, which has been published since the previous CIPS position statement, and to identify possible new recommendations since 2021.

## 2. Materials and Methods

A literature review relating to SRCs in parasports was conducted following a narrative review checklist [14]. Initially, all articles citing the 2021 CIPS position statement were reviewed. Following this, manually scoping the literature for articles published since the 2021 position statement, which included the keyword, was reviewed. Keywords included ‘concussion’, ‘SRC’ or ‘sport-related concussion’, and ‘para’ or ‘parasport’. The databases searched included the Web of Science, SportsDiscus, and ScienceDirect. Articles were excluded if they were not available in English and if they were not original research articles reporting investigation findings; for example, review articles were excluded. Eligible articles were reviewed in-depth, with the relevant data extracted and discussed in this manuscript.

## 3. Results and Discussion

### 3.1. Recommendations for SRC Assessment for Parasport Athletes

#### 3.1.1. Overview and Context

The CIPS consensus statement highlights the importance of clinicians performing regular baseline periodic preparticipation examinations (PPE) on athletes, particularly parasport athletes, to identify a reference point for assessing future symptoms [8]. In addition, it is recommended that the attending clinician should ideally have a thorough understanding of the individual parasport athlete’s history and pre-injury functioning to support an accurate SRC diagnosis. A key emphasis of the statement is that each parasport athlete should receive an individualised SRC management process. The main practical aspect of this may be an issue in certain scenarios, for example, when an athlete or group of athletes travel to an event without their clinician. This may derail SRC management for that athlete or group of athletes. In a recent narrative review of SRC incidence, which was based on vision-impaired (VI) parasports, Teodoro et al. [5] found the number of SRCs sustained by VI athletes to be significantly higher than that of other parasport athletes. On a practical level, this finding suggests that additional safety precautions should be taken during VI parasport training sessions and events. This may take the form of more readily available high-quality protective equipment or the compulsory attendance of an athlete’s/team’s personal clinician. The presence of a personal clinician, who would be knowledgeable about the athlete’s career and sports injury history is viewed as central to a comprehensive SRC management program.

Focusing specifically on falls (a common cause of SRC in parasport) in wheelchair basketball and rugby, Fukui et al. [15] reported that male wheelchair basketball had the highest fall incidence rate at the Tokyo 2020 Paralympic Games. This was followed by female wheelchair basketball and wheelchair rugby, respectively. Regarding the positioning of players, low-pointers in male wheelchair basketball experienced the highest occurrence of falls. Similarly, a literature review by Sá et al. [11] revealed that the head is invariably the most injured region of the body in wheelchair basketball. While parasport training and events, in general, may require additional monitoring for SRC in comparison to their non-parasport counterparts, these findings offer guidance on which specific parasports and sport types warrant the highest level of monitoring.

This CIPS position statement notes that current SRC recognition and assessment tools, such as the SCAT 6 and Sports Concussion Office Assessment Tool 6 (SCOAT 6), assume an athlete to have the ability to see, hear, and read and to have ‘normal’ baseline functioning (such as speech, language, and manual dexterity). Though the SCAT 5 has been proposed for assessing SRC in athletes with limb deficiencies, it is not suitable for those with VI [7]. However, these tools have not yet been fully modified or validated for use across all parasport athletes [16,17]. Acknowledging the impact of various impairment types, a traffic light system was developed with colours representing the potential additional considerations for assessing SRC in parasport athletes. In this system, green represents no additional considerations needed for parasport athletes based on their specific impairment, yellow represents some considerations that may be needed, and red indicates that the specific test should not be used in the parasport athlete SRC assessment. Though a wide range of impairments exist with varying SRC risk, athletes with vision impairments have reported a significantly higher incidence of SRC [1,18].

In their review of SRC incidence rates, Jewell et al. [19] found that approximately 62% of para-SRCs were among those participating in a blind sport. It has previously been reported that these athletes may possibly misunderstand the experience of having a SRC, as their symptoms can differ from those of non-para-athletes. For example, blind footballers who have had a SRC experienced an increase in their sleep disturbance and spatial disorientation compared to non-para-athletes. As these outcome measures do not exist in the current assessment tools and differ from the SRC experiences of sighted people, blind athletes are at an increased risk of their SRC being accidentally misdiagnosed or overlooked [8]. Though there is a sparsity of research assessing the validity of SRC assessment tools across the various impairments in parasport athletes, the high incidence and unique experience of SRC among blind athletes suggests their cohort may be a priority for immediate or upcoming research.

#### 3.1.2. SRC Management for Parasport Athletes

As has been established, an awareness of and having a comprehensive understanding of an individual’s unique experience of a SRC is critical for optimal SRC management and long-term support. Though the CIPS consensus statement recognises the necessity of adequate rest and recovery to lessen the likelihood or severity of SRC symptoms, recent research has found that, in general, parasport athletes experience greater total symptom and severity scores [12]. Additionally, parasport athletes with a history of SRC have reported greater vestibular ocular motor screening (VOMS) effects, including impaired balance, vision, and movement, than those without a history of SRC [20]. Symptoms may also vary based on the athlete’s impairment type. For example, blind footballers experience increased sleep disturbances and spatial orientation post-SRC [8]. Understanding these unique experiences is essential for clinicians to develop and implement practical, effective counteractive approaches.

#### 3.1.3. Return to Learning Following SRC for Parasport Athletes

A return to learning (RTL) post-SRC involves the seamless return to either work or education after injury and the supports in place to facilitate it. Implications of experiencing a SRC can be long-lasting and impede one’s ability to RTL [21]. SRC symptoms that persist for a duration of over three months after the initial injury are recognised as ‘post-concussion syndrome’ (PCS) and can place a significant burden on individuals’ daily living [22]. The 2021 position statement recommends that, as with the general population, parasport athletes should aim to RTL prior to RTS. Any accommodations made in the learning or working environment to support this process should remain in place for longer than for the general population to avoid any unfamiliar changes that could cause stress. This RTL should follow a stepwise approach, with symptoms regularly assessed and movement onto the next phase only resumed when the individual is symptom-free. However, these guidelines fail to account for the variation between able-bodied and parasport athletes’ experiences of SRC and its associated symptoms. For parasport athletes specifically, measuring such symptoms will require an individualised approach, depending on the individual and their abilities. Research investigating this RTL process for parasport athletes is scarce but suggests a range of barriers exist. These include but are not limited to poor mental health support, a lack of understanding from others, and difficult administrative processes relating to obtaining support [23].

#### 3.1.4. RTS Following SRC for Parasport Athletes

The current RTS guidelines for parasport athletes following a SRC recommend that decisions be made on a case-by-case basis given the unique presentation and severity of symptoms throughout the various impairment types. The most recent CIPS position statement highlights that, due to the increased SRC risk in VI sports, these athletes should conduct their full contact practice as a part of their RTS protocol away from other athletes, with this final RTS decision made by a doctor with sufficient knowledge of the specific situation. Research published since this position statement further supports this protocol, as VI athletes continue to be at a higher risk for SRC than sighted athletes [12]. Regarding gender differences in time-off following a SRC, Heron et al. [24] found no significantly significant difference to suggest this variable must be considered throughout SRC management and RTS.

#### 3.1.5. The “3E’s” of Injury Prevention

The 2021 position statement highlights the applicability of the 3E’s of injury prevention to SRC in parasport. These comprise education, enforcement, and engineering (or environment). The former, education, encapsulates teaching clinicians, relevant staff, and athletes about SRCs, specifically in parasports. Of particular importance is ensuring that this information is accessible in athlete-friendly formats, such as offering braille for those with VI. Following this, enforcement related to the rules of a sport and how they may be changed to enhance athlete safety must be considered. For example, the International Federation of Cerebral Palsy Football (IFCPF) introduced a concussion substitution rule in 2020, which aims at allowing ample time for a player to be assessed for a SRC while another takes their place on the field, removing the pressure on medical staff to make quick diagnostic decisions [25]. Finally, engineering or environment relates to the role of equipment in protecting and preventing a SRC. This requires significantly more research before specific recommendations can be developed, with evidence suggesting that the under- and misreporting of SRCs remain a prominent issue in parasport [26].

## 4. Conclusions

The 2021 CIPS position statement on concussion in parasport is a seminal piece of work, supporting and guiding clinicians for their optimal assessment, management, and RTS protocol of para-athletes. However, it lacks guidelines for supporting para-athletes in their RTL. The current guidelines for parasport athlete RTL are identical to those for able-bodied athletes. However, understanding parasport athletes’ unique experience of SRC symptoms and its subsequent effects, such as its impact on mental health, is critical for ensuring successful recovery and seamless RTL. Regarding specific para-sports, men’s wheelchair basketball has the highest incidence rate, based on findings from research at the Tokyo 2020 games, followed by women’s wheelchair basketball and wheelchair rugby, with the head being among the most affected areas of the body.

It remains clear that VI athletes are at an increased risk of sustaining and having unique experiences of a SRC, accompanied by a disruption in spatial orientation and disturbances in their sleep. Moreover, parasport athletes with a history of concussion have reported experiencing greater VOMS effects than their non-para-athlete counterparts. Such symptoms are not currently acknowledged in SRC assessment methodologies, leaving VI athletes at risk of their SRCs being overlooked and undiagnosed. Understanding the individuality of parasport athletes’ SRC experiences will also be vital as they RTL. This is an area that has been heavily overlooked in the past and will require individualised approaches with adequate accommodations available, both short- and long-term. In the future, regular baseline periodic preparticipation examinations, whereby athletes’ baseline functioning is assessed, should be employed. Additionally, assessment tools should be researched and developed to a point where they are easily adaptable to suit athletes with various impairments. Additionally, the ‘3 E’s of injury prevention’ should be considered throughout the development, testing, and implementation of various policies and procedures to promote athlete safety through relevant education, protective equipment, and supportive environments.

## Data Availability

The original contributions presented in this study are included in the article; further inquiries can be directed to the corresponding author.

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
