# Peer review of "Concussion in Parasport: A Narrative Review of Research Published since the Concussion in Para Sport (CIPS) Group Statement (2021)"

_healthcare, 2024, doi:10.3390/healthcare12161562_

Round 1

Reviewer 1 Report

Comments and Suggestions for Authors

Dear authors,

The idea of writing this literature review is a good one. However, it is a narrative review and should be written following specific methodological criteria.

In addition, the literature references seem to be few.

I recommend that you rewrite the review following the checklist that can be found here: https://cdn.elsevier.com/promis_misc/ANDJ%20Narrative%20Review%20Checklist.pdf

https://arts.units.it/retrieve/5abb6ecd-ba09-4a33-9ff6-ab8000402fc6/jcm-2839175-supplementary.pdf

Comments on the Quality of English Language

None.

Author Response

Comments 1 - 

The idea of writing this literature review is a good one. However, it is a narrative review and should be written following specific methodological criteria. In addition, the literature references seem to be few.

I recommend that you rewrite the review following the checklist that can be found here: https://cdn.elsevier.com/promis_misc/ANDJ%20Narrative%20Review%20Checklist.pdf

https://arts.units.it/retrieve/5abb6ecd-ba09-4a33-9ff6-ab8000402fc6/jcm-2839175-supplementary.pdf

Response 1 - 

Sincere thanks for these insightful comments; as authors, we took your suggestion on board and have now edited the article per the checklists which you kindly provided. We have added additional references to the article and these additions are what we consider to be suitable for the current article.

Reviewer 2 Report

Comments and Suggestions for Authors

The paper uses various terms interchangeably (e.g., SRC, concussion). Consistency in terminology should be maintained throughout the document.

Title: The title could be more specific regarding the scope of the paper. It currently does not clearly convey the focus on updates to current practices since the 2021 CIPS statement.

Abstract: The abstract lacks sufficient detail about the methods used and the key findings. It should provide a succinct summary of the objectives, methods, results, and conclusions.

Introduction: The introduction does not provide enough background on the significance of SRC in parasport. More context about the prevalence and impact of SRC in parasport is needed.

Conclusions:

The conclusions do not provide specific actionable recommendations. There should be a clearer link between the findings and the proposed changes to practices or guidelines.

The paper lacks a section on limitations and suggestions for future research, which is crucial for an academic paper.

There are several typos and grammatical errors throughout the paper. Proofreading is necessary to correct these issues.

Line 16-17: "Since 2021,twelve (n =12) articles have emerged..." – There is a missing space after the comma.

Line 20-21: "Current SRC assessment methodologies are not fully inclusive..." – This sentence could be clearer. Consider rephrasing for better readability.

Comments on the Quality of English Language

moderate editing

Author Response

Comments 1 - The paper uses various terms interchangeably (e.g., SRC, concussion). Consistency in terminology should be maintained throughout the document.

Response 1 - Sincere thanks for your comments in this respect; we have now reviewed and edited all terminology to ensure consistency.

Comments 2 - The title could be more specific regarding the scope of the paper. It currently does not clearly convey the focus on updates to current practices since the 2021 CIPS statement.

Response 2 - Thanks for this helpful suggestion, we have now amended the title to reflect the contents of the article

Comments 3 - Abstract: The abstract lacks sufficient detail about the methods used and the key findings. It should provide a succinct summary of the objectives, methods, results, and conclusions

Response 3 - Thanks for this observation, we were constrained by the author guidelines. However we have attempted to remain balanced by aligning our article with the author guidelines and have removed the titles in the abstract. As the word count is naturally limited, we have added more information where possible within the constraints. 

Comments 4 - Introduction: The introduction does not provide enough background on the significance of SRC in parasport. More context about the prevalence and impact of SRC in parasport is needed

Response 4 - Thanks for the comment, we have now added additional information on SRC prevalence, incidence proportion and rate has been added to the text. 

Comments 5The conclusions do not provide specific actionable recommendations. There should be a clearer link between the findings and the proposed changes to practices or guidelines.

Response 5 - Thanks for this comment, we have now included specific actionable recommendations in this section – ‘In future, regular baseline periodic preparticipation examinations, whereby athletes baseline functioning is assessed, should be regularly employed. Additionally, assessment tools should be researched and developed to a point where they are easily adaptable to suit athletes with various impairments.’.

Comments 6The paper lacks a section on limitations and suggestions for future research, which is crucial for an academic paper.

Response 6 - Thanks for this comments, we are aware of these points and as such we have included this information is in the 'results and discussion' and 'conclusion' sections.

Comments 7There are several typos and grammatical errors throughout the paper. Proofreading is necessary to correct these issues.

Response 7 - Many thanks for highlighting these errors, we have now amended and updated the manuscript 

Comments 8Line 16-17: "Since 2021,twelve (n =12) articles have emerged..." – There is a missing space after the comma.

Response 8 - This has now been amended in the manuscript 

Comments 9 Line 20-21: "Current SRC assessment methodologies are not fully inclusive..." – This sentence could be clearer. Consider rephrasing for better readability.

Response 9 - Thanks for suggesting this amendment, this has now been rephrased 

Reviewer 3 Report

Comments and Suggestions for Authors

Overall, this is an important topic. The need for more research regarding concussion in parasport athletes was a clear focus at the most recent international sport concussion consensus conference in Amsterdam in 2022. This review is a nice idea to collate the recent literature on concussion in parasport athletes.

Introduction: 

In general, good content and organization.

Materials and Methods:

The manuscript would benefit from a bit more detail, so that the methods could easily be reproduced by the reader. At the very least, I would recommend adding the exact keywords that were used and which databases were searched. A statement on if any manuscripts were excluded after the initial search would also be useful.

Results and Discussion:

This section is well organized by several key areas to sport-related concussion in general, including assessment, management, return-to-learn/work, return-to-sport, and prevention. One area that is missing that I believe should be added to the manuscript is the use of exercise in the management of concussion. There has been a significant amount of research into this in recent years, with large changes in the management of rest and physical activity, e.g. subsymptom threshold exercise, particularly by John Leddy and his group at University at Buffalo. Even if there is not any parasport-specific research to include in this review, I think the topic should be mentioned as a discussion point, which could include a current dirth of research, as well as discussion of how the use of exercise in general athletes after concussion may have significant limitations in translation to parasport athletes (eg the Buffalo Concussion Treadmill Test may not be feasible for some parasport athletes).

You may even consider an additional section for key areas of post-concussion evaluation and management where there is currently little to no research, which would include the use of exercise.

Conclusion: 

The conclusion can be pared down to more neatly summarize the key concepts that were revealed by the review. As is, the conclusion has too much specific retelling of the prior section (Results and Discussion)

Comments on the Quality of English Language

Overall easy to follow as a primary English speaker. The manuscript is in need of some general grammatical edits throughout, some of which stand out more prominently. For example, in the introduction, lines 62-64:

"Since this position statement was published in 2021, many research articles have emerged to suggest. recommendations and guidelines which are current."

Author Response

Comments 1 - Overall, this is an important topic. The need for more research regarding concussion in parasport athletes was a clear focus at the most recent international sport concussion consensus conference in Amsterdam in 2022. This review is a nice idea to collate the recent literature on concussion in parasport athletes.

Introduction: In general, good content and organization.

Materials and Methods: The manuscript would benefit from a bit more detail, so that the methods could easily be reproduced by the reader. At the very least, I would recommend adding the exact keywords that were used and which databases were searched. A statement on if any manuscripts were excluded after the initial search would also be useful.

Response 1 - Sincere thanks for your comments on our work, we have now updated the text with information on keywords and exclusion criteria. 

Comments 2Results and Discussion: This section is well organized by several key areas to sport-related concussion in general, including assessment, management, return-to-learn/work, return-to-sport, and prevention. One area that is missing that I believe should be added to the manuscript is the use of exercise in the management of concussion. There has been a significant amount of research into this in recent years, with large changes in the management of rest and physical activity, e.g. subsymptom threshold exercise, particularly by John Leddy and his group at University at Buffalo. Even if there is not any parasport-specific research to include in this review, I think the topic should be mentioned as a discussion point, which could include a current dirth of research, as well as discussion of how the use of exercise in general athletes after concussion may have significant limitations in translation to parasport athletes (eg the Buffalo Concussion Treadmill Test may not be feasible for some parasport athletes). You may even consider an additional section for key areas of post-concussion evaluation and management where there is currently little to no research, which would include the use of exercise.

Response 2 - thank you for the insightful comments, we have now mentioned this in the 'introduction' section of the article. Please note that we did not discuss it in detail as it did not appear in the manuscripts/articles which emerged as new research from our literature searches.

Comments 3 Conclusion: The conclusion can be pared down to more neatly summarize the key concepts that were revealed by the review. As is, the conclusion has too much specific retelling of the prior section (Results and Discussion)

Response 3 - Sincere thanks for this suggestion, we have now reviewed the content and shortened the conclusion to present it in a more concise manner. 

Comments 4Comments on the Quality of English Language - Overall easy to follow as a primary English speaker. The manuscript is in need of some general grammatical edits throughout, some of which stand out more prominently. For example, in the introduction, lines 62-64:

"Since this position statement was published in 2021, many research articles have emerged to suggest. recommendations and guidelines which are current."

Response 4 - Thanks for this comment, we have fully reviewed and edited the article in accordance with your suggestion. 

Round 2

Reviewer 1 Report

Comments and Suggestions for Authors

Dear authors,

thank you so much for your replies. The manuscript is much improved.

However, I recommend that you put the following sentence at the beginning of the “Materials and Methods” section:

A literature review relating to SRC in parasports was conducted following the Narrative Review checklist [insert reference number]. 

  1. Add reference:
  2. Narrative Review Checklist. Available online: https://www.elsevier.com/__data/promis_misc/ANDJ%20Narrative%20Review%20Checklist.pdf (accessed on 14 July 2024).
  1.  
  2. Thank you!
Comments on the Quality of English Language

-

Author Response

Comments 1 - Dear authors, thank you so much for your replies. The manuscript is much improved.

However, I recommend that you put the following sentence at the beginning of the “Materials and Methods” section: A literature review relating to SRC in parasports was conducted following the Narrative Review checklist [insert reference number]. 

  1. Add reference:
  2. Narrative Review Checklist. Available online: https://www.elsevier.com/__data/promis_misc/ANDJ%20Narrative%20Review%20Checklist.pdf (accessed on 14 July 2024).

Thank you!

Response 1 - Thanks for your kind comments; we have now edited your suggested comment into the manuscript and updated the reference list for completeness. 

Reviewer 2 Report

Comments and Suggestions for Authors

It is ready to be published.

Author Response

Response 1 - Thanks for your kind comments; we appreciate your suggested edits and the time you took to review this work. 

Reviewer 3 Report

Comments and Suggestions for Authors

I am satisfied with the edits made and feel they have significantly improved the manuscript. I believe the manuscript is now acceptable for publication.

Author Response

(The authors gave the same response as above.)
